# Understanding ADHD: Toward an Innovative Therapeutic Intervention

**DOI:** 10.3390/bioengineering8050056

**Published:** 2021-05-01

**Authors:** Allyson Camp, Amanda Pastrano, Valeria Gomez, Kathleen Stephenson, William Delatte, Brianna Perez, Hunter Syas, Anthony Guiseppi-Elie

**Affiliations:** 1Center for Bioelectronics, Biosensors and Biochips (C3B^®^), Department of Biomedical Engineering, Texas A&M University, College Station, TX 77843, USA; allysoncamp@tamu.edu (A.C.); apastrano@tamu.edu (A.P.); val_go@tamu.edu (V.G.); kathleen_katie@tamu.edu (K.S.); wedelatte19@tamu.edu (W.D.); bri_2024@tamu.edu (B.P.); swimchem16@tamu.edu (H.S.); 2Houston Methodist Institute for Academic Medicine and Houston Methodist Research Institute, 6670 Bertner Ave., Houston, TX 77030, USA; 3Department of Electrical and Computer Engineering, College of Engineering, Anderson University, Anderson, SC 29621, USA; 4ABTECH Scientific, Inc., Biotechnology Research Park, 800 East Leigh Street, Richmond, VA 23219, USA

**Keywords:** ADHD, neurostimulation, therapy, design, adherence

## Abstract

Attention deficit hyperactivity disorder (ADHD) is a pervasive condition affecting persons across all age groups, although it is primarily diagnosed in children. This neurological condition affects behavior, learning, and social adjustment and requires specific symptomatic criteria to be fulfilled for diagnosis. ADHD may be treated with a combination of psychological or psychiatric therapeutic interventions, but it often goes unattended. People with ADHD face societal bias challenges that impact how they manage the disorder and how they view themselves. This paper summarizes the present state of understanding of this disorder, with particular attention to early diagnosis and innovative therapeutic intervention. Contemporary understanding of the mind–brain duality allows for innovative therapeutic interventions based on neurological stimulation. This paper introduces the concept of neurostimulation as a therapeutic intervention for ADHD and poses the question of the relationship between patient adherence to self-administered therapy and the aesthetic design features of the neurostimulation device. By fabricating devices that go beyond safety and efficacy to embrace the aesthetic preferences of the patient, it is proposed that there will be improvements in patient adherence to a device intended to address ADHD.

## 1. Introduction

Attention deficit hyperactivity disorder (ADHD) is a neuropsychiatric disorder present in as much as 5% of the population [1]. According to the American Psychiatric Association, the disorder manifests in a pattern of inattention and/or hyperactivity–impulsivity that interferes with the completion of daily tasks, school, and work [2]. The syndromic hyperactivity, impulsivity, and inattention result in three types of ADHD categorized as primarily inattentive, primarily hyperactive–impulsive, and a subtype combination of the former two [3]. The disorder is diagnosed by a clinical professional who determines if certain clinical criteria are met. ADHD is primarily treated with stimulant medications, behavioral therapy, or a combination of both. A nationally representative survey estimated that at least 4.4% of U.S. adults have ADHD, but less than 10.9% of those patients receive pharmacological or non-pharmacological therapeutic interventions [4]. It is further estimated that 6.1 million children in the U.S. have ADHD, yet as many as 23% may not receive any medication or behavioral treatment [5]. Individuals affected by the disorder carry an additional burden from societal bias caused by a widespread lack of understanding of the condition and its treatments. Breakthroughs in alternative therapeutic interventions follow a growing understanding of the etiology and progress of the condition. Currently, biomedical devices are being developed to address the symptoms of ADHD in a radically new way. Recently, the U.S. Food and Drug administration provided pre-market approval for a prescription-only neurostimulation device as a therapeutic intervention. This approval came following the success of their baselined, unblinded study that met the FDA’s criteria. The Monarch external Trigeminal Nerve Stimulation (eTNS) System delivers low-level electrical pulses to parts of the brain associated with ADHD [6]. These pulses increase activity in areas of the brain that regulate emotion, attention, and behavior [7]. Understanding all aspects of ADHD, from the associated basis in heredity through the environmental/exposure factors to the neurological manifestations among various demographics, is critical to ensuring appropriate therapeutic intervention given this paradigm shift in therapy.

## 2. Pathophysiology and Causes

ADHD is a neuropsychiatric disorder known to be highly heritable and linked to a variety of genetic factors that affect neurological and molecular systems [8]. Epigenetic contributions and prenatal chemical exposures may contribute to the etiology of ADHD; however, overwhelming evidence in the literature indicates that genetics is the main causal influence of ADHD [1]. Studies reveal anywhere from 70% to 80% heritability of ADHD among adolescents [9,10]. Heritability estimates of ADHD in adults are notably lower at 30% to 40% [9]. This dramatic difference can be attributed to errors in the measurement of symptoms and should neither discredit the heritability rates observed in children nor suggest a strong preponderance that ADHD may be acquired. Sibling and twin studies are often used when searching for a genetic link in disease [11]. The risk of being diagnosed with ADHD is up to 13 times higher for the younger siblings of children who have previously been diagnosed with the disorder than for those who do not have the disorder [12]. These data are confirmatory of the existence of genes with a causal correlation for ADHD susceptibility [13]. There is convincing evidence for genome-wide significance in 12 independent loci, specific locations in the human genome, in which DNA variations are robustly associated with the risk for ADHD [14].

Adhesion G Protein-Coupled Receptor L3 (ADGRL3) (previously known as LPHN3) is a gene associated with ADHD in a large sample of children and adults [9]. ADGRL3 moderates neurotransmitter release in the brain and might contribute to brain development [15,16]. Suppressing ADGRL3 in mice and the homologous ADGRL3.1 (previously LPHN3.1) in zebrafish causes hyperactivity. Decreased ADGRL3.1 activity impacts the function of dopaminergic neurons, which regulate locomotion in humans, often resulting in the hyperactive and impulsive symptoms commonly observed in ADHD patients. ADGRL3 interacts with other genes that may increase susceptibility to ADHD. There are multiple genes spanning a section on chromosome 11 that interact with ADGRL3. The interactions of single nucleotide polymorphisms (SNPs) in the 11q cluster with SNPs of ADGRL3 doubled the risk of ADHD and increased the severity of the disorder [10,17]. The genes in the cluster are important to brain development, establishing once again the neurological impact associated with ADHD.

A central theory in the pathophysiological causes of ADHD is rooted in delays of the reward system in executive functions, leading to a reduction in control and inhibitory deficits [18]. The executive circuit that links the prefrontal cortex to the dorsal neostriatum, which is involved in dopamine release, is believed to be disrupted in patients with ADHD, as identified in imaging studies comparing active regions of the brain [18]. This executive dysfunction may be derived from genetics. Other tracts that are believed to be abnormal in patients with ADHD are the basal ganglia to the dorsomedial thalamus and the thalamus to the prefrontal cortex. Both structural and functional neuroimaging studies have shown differences in these circuits compared to control groups [19]. These abnormal patterns, revealed by imaging studies regarding the differences in executive function circuits and the reward system areas of the brain between controls and ADHD patients, have become the basis for neurostimulation technology and its application to ADHD. A newly developed adaptive frequency-based model of whole-brain oscillations relative to that of the resting-state fMRI data of healthy controls revealed two ADHD sub-groups, consistent with the distinct behavioral phenotypes (i.e., depression and hypomanic personality traits). This allowed for the stratification of patients for neurostimulation therapy and potential personalization of this therapeutic intervention [20].

Prenatal exposure to chemicals (medicines, drugs, and pollutants) in the environment has also been shown to increase the risk of having the disorder. Antiepileptic drugs, such as valproate, an anticonvulsant and mood-stabilizing drug for the treatment of seizures and manic episodes associated with bipolar disorder, and carbamazepine, an anticonvulsant, are used to treat neurological disorders, but their use in pregnant mothers heightens the risk of congenital neurodevelopmental problems in their children [21,22]. A Danish valproate study revealed nearly a 50% increase in ADHD risk in children exposed to the drug before birth [23]. Lead is also connected to ADHD vulnerability [24]. This metal is known to be toxic and can affect a developing child’s nervous system. In both high and low doses, lead contamination positively correlates with ADHD [25]. Understanding the role of contributing external factors is both challenging and vitally important because they represent a point of leverage in building awareness for reducing exposure and hence the risk for the condition. Studies are continuing to explore novel genetic connections and environmental factors that may alter neurological pathways in children and lead to hyperactivity.

## 3. Clinical Manifestations

The three subtypes of ADHD currently recognized are predominantly inattentive (IA); predominantly hyperactive-impulsive (HI); and the “combined” type, a blend of the other two, which is the most common [26]. Clinicians use diagnostic criteria from the Diagnostic and Statistical Manual of Mental Disorders (DSM-5) and the International Statistical Classification of Diseases and Related Health Problems (ICD-10) manuals to evaluate patients for a proper ADHD diagnosis [27]. This diagnosis includes reviewing symptoms present in home and school settings from parent, teacher, and patient perspectives through questionnaires; interviews; and reviewing records, such as academic transcripts. Additionally, family history is reviewed, and physical examinations are completed for the patient [28]. Clinical manifestations and symptoms must be present in several settings such as in a home, school, or work environment. At least six of the following clinical manifestations must be observed and documented for the diagnosis of the IA subtype in a child of up to 16 years old: difficulty with paying attention to detail resulting in careless mistakes, difficulty maintaining focus, a pattern of losing necessary objects, appearing not to listen when addressed, displaying difficulty with following through on assignments or tasks, hardship in organizing activities or personal possessions, and repeating avoidance of tasks that demand intellectual effort [2]. Only five symptoms are required for the diagnosis of adolescents older than 17 years or adults. This subtype is defined by other symptoms, such as being distracted or unfocused, and is often diagnosed in older adolescents [2]. Female patients are more commonly diagnosed with the IA ADHD subtype [28]. The HI subtype requires the consistent presentation of clinical manifestations, including restlessness, fidgeting, constant interruption of others, problems with remaining still during activities, blurting out answers, hardship in staying quiet, restlessness, over-energetic activity, an “on-the-go” quality, and difficulty staying on task, all of which commonly demonstrate impulsive and hyperactive behavior for at least six months prior to diagnosis [2]. The third subtype manifests with a mixture of the aforementioned symptoms that may vary between patients and an overall difficulty with impulsiveness, hyperactivity, and inattention. Other symptoms related to inattentiveness for elementary-aged patients could include forgetting a backpack, forgetting assignments, overlooking details, daydreaming, or organizational difficulty, which makes the completion of assignments more difficult and could hinder academic performance and skills. Adults can also be affected by similar symptoms, such as forgetfulness, only in a workplace or home setting rather than at school. Symptoms related to hyperactivity or impulsivity include difficulty following classroom etiquette, excessive talking, and other related symptoms, making it more challenging for the patient to stay on task [29].

These associated symptoms disrupt the academic and social aspects of a patient’s life. ADHD symptoms related to behavior can greatly hinder the social abilities of patients, making it harder to grasp social cues, which could pose difficulty in forming friendships with peers. Patients with HI ADHD may engage in spontaneous and risky behavior that can make them a danger to themselves [29]. The three subtypes each contribute potential academic obstacles and can sometimes be accompanied by learning disorders leading to increased difficulty studying, paying attention in class, and making careless mistakes on assignments. Anywhere from 31% to 45% of students with ADHD are affected by the co-occurrence of learning disabilities, presenting further academic barriers in math, writing, and reading, in addition to ADHD symptoms [30]. The societal misunderstanding of ADHD can lead young patients to be mislabeled as misbehaving or careless students. Some comorbidities arise from the stress and frustration that emerge as patients acknowledge their behavioral differences and struggle to conform to a social setting in order to avoid social repercussions or academic consequences for their outbursts [31]. For example, difficulty with social skills can lead to higher risks for disorders such as anxiety or depression [32]. Children with ADHD can suffer from a certain degree of emotional impairment, with difficulty controlling feelings including anger or aggression [29]. These difficulties can commonly be accompanied by disorders such as oppositional defiant disorder and conduct disorder, in which patients exhibit strong patterns of repeated outbursts, acts of defiance, hostility, and aggression toward individuals [29]. Research indicates that over 1/3 of all ADHD patients have an additional comorbid disorder [28]. This trend is further explored in Figure 1, with the most frequent comorbidities being any mental, emotional, or behavioral disorder, accounting for 64% of the comorbidities noted at diagnosis.

Additionally, ADHD is associated with sleep problems including insomnia, reduced sleep duration, and issues affecting circadian rhythms [33]. Some research has indicated that some ADHD stimulant medications, such as methylphenidate, can impact the regulation of circadian rhythm and worsen the insomnia-like symptoms already present with ADHD [34]. Sleep disorder issues for ADHD patients can also be caused by the presentation of coexisting disorders, such as depression or anxiety [33]. Other comorbid disorders observed in ADHD patients include a higher prevalence for mood disorders, autism spectrum disorder, substance use disorder, and bipolar disorder [35]. Research has shown that the proper treatment and management of both disorders can help improve symptoms [35].

A further characteristic of ADHD pertains to the self-defeating behavior known as emotional dysregulation [36]. This behavior is generalized as a group of behaviors or a single behavior that leads to damage or defeat of the patient’s well-being and interests [36]. This is an intersectional phenomenon that is not exclusive to ADHD patients, but can often characterize many of the most observable ADHD symptoms that manifest themselves in more harmful ways, directly degrading the happiness and stability of the patient through self-sabotage. Treatment can be difficult as this overlap can make clinical isolation difficult. However, studies have shown that the same psychostimulants effective in ADHD treatment are in turn effective in the management of emotional dysregulation [36].

## 4. Diagnosis

ADHD is most commonly diagnosed in children and adolescents (ages 5–17). Figure 2 illustrates that according to parent reporting, the age of 12 in boys is the most prevalent diagnosis age, while 14 years or 16 years is the most prevalent in girls. Contrary to previous beliefs, ADHD exists in the adult population, but it presents with fewer symptoms as adults may develop compensatory behavioral adjustments to symptoms [37]. Approximately 8% of children and 4% to 5% of adults have been diagnosed with ADHD [38]. The two main diagnostic systems are the DSM-5 used in the United States and the ICD-10 used globally [39]. In comparison, the ICD-10 has stricter criteria than the DSM-5 as shown in a study with young children where only 26% of those who met the DSM criteria also met the ICD criteria [40]. The DSM-5 criteria are apportioned into inattention and hyperactivity–impulsivity, with ten points distinctly differentiating between the two subtypes. Table 1 shows the prominent symptoms for each subtype. A common rating scale is the ADHD-RS-V which is derived from the DSM-5 criteria outlined in Table 1 and requires a clinical assessment of symptoms for proper diagnosis [2]. These diagnoses fall into the three previously introduced subtypes: primarily inattentive (IA), primarily hyperactive-impulsive (HI), and a combination. The evaluation takes the form of an 18-part questionnaire, split equally between questions about IA- and HI-specific symptoms [41]. There is a home/parent version and a school/teacher version which help assess how adversely children are affected in these two settings [41]. Some of the symptoms the ADHD-RS-V evaluates include difficulty sustaining attention, forgetfulness, distractibility, excessive talking, and restlessness [38]. A diagnosis requires that these symptoms are present for at least 6 months, with at least 6 symptoms being displayed by a child or 5 symptoms being shown by an adult. [38,39,42]. The total score is derived from the 4-point Likert system and converted to a percentile which is assessed based on age and gender to give an official diagnosis. A newer diagnostic test, the Quantified Behavioral (QB) Test, is an improved continuous performance test designed to track the inattention and impulsivity of the adult or child being tested as well as their motion and bodily activity during the test [43]. The results are reported in a weighed matter so that a score beyond 93% would indicate a high likelihood of ADHD [43]. Computerized QB Test Plus (Qb+) emphasizes objectivity, allows for remote diagnoses, and accommodates independent partitioning among the core symptoms of hyperactivity, inattention, and impulsivity [44,45].

Many other rating systems exist for ADHD. One example is the National Institute for Children’s Health Quality (NICHQ) Vanderbilt Assessment scale which is much longer than the ADHD-RS-V. The NICHQ assessment consists of 55 questions for the parent form and 43 questions for the teacher form that cover both symptoms and performance [46]. The Swanson, Nolan, and Pelham-IV Questionnaire (SNAP-IV) is another option, but it can be used for more than just ADHD. SNAP-IV is a compilation of 90 questions used to gauge the symptoms of all 3 ADHD types, oppositional defiant disorder, and other psychiatric disorders identified by the DSM [47]. The Adult ADHD Self-Report Scale (ASRS) is another assessment that uses criteria from the DSM-5 and consists of only 18 questions [48]. Other clinical assessments include the Gordon Diagnostic System (GDS), a suite of 11 tests, often useful as a clinical component of an ADHD evaluation [49]; the Behavior Rating Inventory of Executive Function (BRIEF) used in the diagnosis of ADHD [50]; the FBB-HKS total score, a German ADHD rating scale, completed by both parents and teachers [51]; and the Wender Utah Rating Scale (WURS), a 61-symptom list rated on a 5-point scale [52]. An alternative scale, the Conners’ Teacher Rating Scale, has many versions all consisting of a 59-, 28-, or 10-item form and then a collection of scales based on a specific behavior, numbering 6, 3, or 2 separate scales [53]. The use of behavior rating scales, symptom validity tests, and cognitive tests further improves the chance of avoiding a misdiagnosis. Since ADHD is a syndrome that deals with psychological and behavioral matters, psychologists, psychiatrists, or neurologists, specifically specializing in children and adolescents, are typically those who make these diagnoses [54].

In addition to these symptom-based tests, a technological method of diagnosis exists that employs electroencephalography (EEG). The FDA approved the Neuropsychiatric EEG-Based Assessment Aid (NEBA) system for supporting a clinical ADHD diagnosis in 2013 [57]. As per FDA regulations, this device is not meant to be used as a stand-alone diagnostic technique, but it is encouraged for supplemental use and to help with treatment plans [58]. The 20 min assessment involves a recording electrode at midline central (CZ) and a ground electrode at midline frontal (FZ) [59]. The NEBA system analyzes these data with an algorithm to identify brain wave ratios and uses high to moderate theta/beta ratios to identify ADHD [59]. Historically, ADHD has been associated with an elevated theta/beta brain wave ratio due to increased theta levels (4–7 Hz) and lower beta levels (14–30 Hz) [60,61]. This ratio is not homogenous among the ADHD population. For example, the IA ADHD subtype is often indicated by an alpha (8–14 Hz) deficit rather than the typical theta/beta ratio [61,62]. This technology is more useful for prognosis and helping with subtype identification [60].

Given the genetic basis of the etiology of ADHD [8], the prospect exists for the eventual genotyping of the disease. However, being a multi-gene syndromic disorder with implications across multiple gene families [14], this may require complete gene sequencing. A key consideration is the proper subtype diagnosis to allow the appropriate therapeutic intervention, i.e., a movement toward theragnostics.

With the diagnosis of ADHD, many patients stand at a greater risk of suffering from other accompanying mental disorders, such as depression or dysthymia (persistent mild depression). Approximately 16% to 37% of patients diagnosed have comorbid major depressive disorder/dysthymia, putting them at a much higher risk of suicidal behavior [63]. Growing evidence suggests that this stems from the negative influences and effects of this disorder, such as deficits in social skills [64], strained family relations [65], declined academic success [66], and lack of financial independence indicators later in life [67]. As ADHD projects all these social hardships onto a single person, it has the potential to greatly decrease their sense of self-worth and self-esteem. This can also leave patients prone to severe substance abuse when compared to their unaffected counterparts [38].

Many persons with ADHD do not seek proper treatment, as several mental health disorders carry a negative stigma that impedes the pathway to therapy. In recent studies, around 25% of parents did not want “their child to make friends with a child with ADHD” with some even disregarding the possibility of the presence of a disorder and blaming the child’s family for their inability to raise a child properly [68]. Similarly, this type of negative stigma can also be seen in children, as children diagnosed with ADHD are overall less favored by their unaffected peers. Behavioral manifestations of ADHD, such as interrupting, are sometimes viewed as character flaws instead of uncontrollable symptoms. This has caused expressions of uncertainty regarding the validity of the diagnostic process and skepticism regarding the efficacy of medications prescribed as treatment [68]. The general negative stereotype ascribed to those diagnosed with ADHD also greatly affects the patient’s treatment/therapy adherence, treatment efficacy, and symptom aggravation as they do not want to be an outlier. However, due to growing awareness of the syndrome, its genetic basis, and the roles of therapeutic intervention, this disorder is becoming less stigmatized and better understood.

## 5. Clinical Management

Clinical management follows three broad paths: behavior adjustment and accommodations, medication, and a combination of both. The standard of care for patients with ADHD varies with age and manifest intensity of the condition. However, according to the National Institute for Health and Care Excellence (NICE) standards, before medication, lifestyle should be the initial focus for adjustments [27]. This includes regulating what the patient eats, verifying the quality of what the patient eats, and making sure that the patient is on an age- and physical condition-appropriate exercise regimen. For very young children, this is the primary treatment aside from training their guardians, due to a common aversion to medicating young children [27]. Properly educating all parties involved in a child’s care is a standard across all variations as misinformation about ADHD is prevalent. Psychological approaches, in the form of group and individual therapy for parents and patients, are recommended along with medication in those with less-inhibiting forms of ADHD, whereas medication is recommended first with more inhibiting forms of ADHD. The three most used and researched medications are methylphenidate, atomoxetine, and dexamphetamine [27]. Each is specific to a set of symptoms and manifestations of ADHD. They come along with their own side effects, typically some sort of small to large behavioral change. This change can manifest as anything from high aggression to more frequent intrusive and suicidal thoughts [27]. These treatment methods are typically recommended by the diagnosing primary physician.

Finding the best-fit medication for a patient is usually carried out through trial and dose adjustments, paying careful attention to contraindications, attenuation of symptoms, or the emergence of comorbidities. Additionally, classified as treatment-emergent adverse effects, standard stimulant symptoms for methylphenidates and amphetamines most often include a combination of the following: insomnia, diminished appetite, headaches, nausea, or gastrointestinal pain. [69]. The following three stimulants have also been observed to cause elevated blood pressure and elevated heart rates for patients. [69]

**Methylphenidate** is a first-line drug in the treatment of ADHD. It stimulates the central nervous system (CNS) by affecting the bioavailability of chemicals in the brain and nerves, such as dopamine and norepinephrine, that contribute to hyperactivity and impulse control. Often taken orally, but sometimes delivered transdermally, it is sold under brand names such as Ritalin, Metadate ER, Concerta, and Methylin.

**Atomoxetine** is a first line drug in the treatment of ADHD. It is a norepinephrine reuptake inhibitor and is a non-stimulating drug believed to work by increasing norepinephrine and dopamine levels in the brain. It is taken orally and may be used with a stimulant. It is sold under the brand name Strattera with generics produced by APOTEX, USA, and Teva Pharma, USA. Specific treatment-emergent adverse effects for atomoxetine include headaches, dry mouth, dizziness, appetite changes, and gastrointestinal issues such as nausea, vomiting, constipation, and abdominal pain. [69].

**Dexamphetamine** is used in the treatment of ADHD. It is an amphetamine that stimulates through the inhibition of the transporter proteins responsive to the monoamine neurotransmitters serotonin, norepinephrine, and dopamine. It is taken orally and is sold under the brand names Dexedrine Spansule, ProCentra, and Zenzedi.

The implementation of stimulant medication as an ADHD treatment has been the standard in clinical settings. The stigma that children’s early usage of controlled substances could pose issues with substance abuse disorders later in life has been researched. Parents have concerns because stimulants, such as methylphenidate, amphetamine, and opioids, can both increase dopamine release, so their concurrent use can reinforce dopamine signals and increase abuse risk [70]. However, a meta-analysis from previously completed longitudinal studies on substance abuse for children who used stimulants for ADHD treatments versus children who did not use stimulants indicated that ADHD stimulant usage as a child did not increase or decrease the likelihood for ADHD patients to later develop substance abuse issues compared to other children [71]. Despite this research, the negative connotation toward stimulant use, as well as the long-term effects on ADHD patients who depend on stimulants for the duration of their lives, still leaves parents hesitant to use this form of treatment. Guanfacine is a non-stimulant medication that can be used to treat children and adolescents who have found stimulant medications to be ineffective or want to avoid their use due to the potential side effects [72].

Through guidance made possible under the Individuals with Disabilities Education Act (IDEA), schools serve as a major resource in the management of patients with ADHD. The diagnosing physician will often refer to proper psychotherapy and pharmacological routes depending on the severity of the patient’s case, which may include accommodations at school [28]. The local school district is instrumental in this process, helping parents create a plan for their child at school. As for the treatment of adolescents with ADHD, treatment methods that are a combination of primary care providers with psychiatrists may not lead to the increased or decreased mitigation of symptoms but have been observed to help the comfort of the patients being treated [73]. Inside and outside of school, treatment plans must be highly individualized, as ADHD’s manifestation and symptoms are quite variable. The individualized education programs (IEPs) for students with ADHD are one example of specific plans made to help students succeed in school [74].

With anywhere from 40% to 80% of diagnosed children having this condition continue into their adolescent development, ADHD symptoms are not eliminated entirely as the patient ages [75]. Its continuation into adulthood can be difficult to discern because ADHD is often comorbid with another behavioral or mood disorder [75]. Patients with ADHD that persists into adulthood struggle with continuing tasks, consistently putting forth effort, and self-motivation [75]. Though treatment can help mitigate the condition’s symptoms, a trend with increased persistence of the disorder and treatment has been observed, with the severity increasing should the patient stop their regimen [76]. Individuals with ADHD extending into adulthood are at higher risk for developing substance use disorders, obesity, antisocial personality disorder, and are more likely to be involved in car crashes [38]. These all often contribute to an overall degradation of the quality of life that can lead to a high rate of death in persistent ADHD cases [76]. Overall, the management of ADHD varies drastically due to stigma and misinformation, as well as how difficult the condition is to diagnose with its broad symptoms and high rate of comorbidities. The constants are that education about ADHD and consistent, positive lifestyle habits are beneficial with all forms of management and treatment.

Neurofeedback treatment is an alternative to medication enabled through the use of an EEG and brain wave training [51]. During treatment, the patient’s theta and beta waves are actively monitored, and the administering professional trains the patient to lower the theta waves and increase the beta waves [51]. This method is proven to have the highest rate of efficacy among the non-pharmaceutical alternatives to the treatment of ADHD, though the particulars of the effects are yet to be studied [51].

The anthroposophical therapy of Rudolf Steiner (1861–1925) and Ita Wegman (1876–1943) has been applied to ADHD and has received mixed reviews, specifically for weakness in complying with the tenets of evidence-based medicine [77]. This holistic approach includes treatment modalities, such as art therapy, rhythmical massage therapy, game therapy, and eurythmy or movement therapy [78]. Although the study performed is empirically weak due to the limited participants in a non-control group design, its positive outcomes indicate the need for further research to verify the medical efficacy of such therapeutic approaches [77].

## 6. Societal Impact

ADHD is proven to have adverse effects on patients, including hyperactivity, inattentiveness, learning disabilities, emotional imbalance, and depression or anxiety [79]. In addition to the patient’s personal challenges with ADHD, they face stereotypes and societal bias which are destructive to their self-esteem and confidence. From peer teasing and social neglect to differences in gender stereotypes, people with the disorder constantly face adversity from society [80]. Sources indicate that these tunneled perspectives of society perceive individuals with ADHD as forgetful, lazy, messy, disobedient, “improper”, and outcasts to the societal norm [80,81]. While it is easy to focus on the burden resulting from the disorder and inaccurate perceptions, it is important to remember that many individuals diagnosed with ADHD possess strengths that can act as long-term advantages. Qualities such as creativity, resilience, risk-taking, grit, and the ability to quickly move from one task to another can greatly benefit an individual with ADHD. These characteristics are altogether highly attractive and useful to an evolving society [82].

Among people who have been diagnosed with ADHD, there are discrepancies in how they are viewed by society based on their age and sexual identity, the status of their medical treatment, and their economic standings [80,83,84]. Women, minorities, and individuals that have historically been regarded as lower in society are the most susceptible to being treated harshly or shamed by others [79,80]. Gender and racial equality are very prominent in determining the treatment of those with ADHD. In general, women with ADHD suffer more from anxiety, alcohol abuse, and drug abuse, while in comparison, men experience hyperactivity and abrupt behavior which results in a more diagnosable condition [79]. Similarly, a research study investigating the prevalence of ADHD in Arabian society examined a cross-sectional cohort group of primary school children aged 6–12 in Qatar and ultimately indicated supportive information of ADHD symptoms in approximately 9.4% of school-aged children in the cohort [81]. In this case, the lack of technology, statistics, and medical treatment in these regions inhibited logical interpretations of ADHD, causing such regions to devalue individuals based on family conflicts and biology. From an economic standpoint, statistics show that a proportion of 4.8% of adults with ADHD was more likely to experience unemployment and, therefore, contribute less to the national economy [83]. Some sources suggest that unethical employer decisions and workplace conditions prevent a fair advantage for individuals with ADHD to sustain stable careers [81,83,85]. These ethical and cultural gaps along with multiple other societal factors discourage those who might have ADHD from seeking medical assistance early on or even revealing their disorder to the public after having been diagnosed.

Despite the stereotypes and arguably unethical claims that society associates with this condition, there are some incredibly positive characteristics that researchers have discovered in individuals with ADHD. In fact, author Dale Archer emphasizes how patients diagnosed with ADHD are often highly successful entrepreneurs [82]. Some individuals with ADHD are clinically proven to exhibit extraordinary creativity, hyperfocus, multitasking, energy, determination, and resilience when facing adversity [82]. This combination of skills leads people with the disorder to be natural innovators [82]. Individuals who overcome the personal challenges of ADHD, with or without medical assistance, may become more appreciative towards their unique attributes and concentrate more on productive strengths to help them to achieve their aspirations [82,86].

## 7. Innovations in Therapeutic Intervention

Improved understanding in the molecular etiology of ADHD has created opportunities for technological interventions based on neurostimulation. Neurostimulation is the purposeful modulation of the nervous system’s activity using invasive (e.g., deep brain stimulating microelectrodes) and non-invasive (e.g., transcranial magnetic stimulation (TMS or rTMS) or transcranial electric stimulation, tES, such as transcranial direct current stimulation (tDCS) and transcranial alternating current stimulation (tACS)). The success of neurostimulation can likely be attributed to the “electroceutical theory” of neurostimulation and the “augmentation theory” of neurostimulation. Electroceuticals are nascently produced biochemicals whose levels are potentiated under the influence of the applied electric field [87]. The upward or downward changes in the chemical potential of one or multiple biochemicals alter the activity of communication between specific nerve fibers to achieve therapeutic effects. The augmentation theory purports that the therapeutic benefits arise from physicochemical means, such as changes to the transmembrane potentials, membrane permeability, or electroactivity of receptors or receptands, under the influence of the applied electric field [88,89].

The techniques of neurostimulation vary widely, reflecting the diverse possible clinical indications. However, studying this treatment and its direct effects is a challenge due to the inability to eliminate confounding factors such as brain maturation and working with a population that is overall forgetful of tasks and defiant [19]. Depending on the condition being treated and the therapeutic outcome sought, neurostimulation can be used to target specific regions of the brain, the whole brain, or particular neurochemical pathways of a specified area of the brain [90]. Transcranial magnetic stimulation (TMS) works to hyperpolarize and depolarize the neurons through magnetic pulses delivered through a coil placed on the patient’s scalp. Changing the magnetic field type affects which area of the brain is targeted via magnetic focusing. Repeated low frequency stimulation (<5 Hz) created a long-term reduction in the excitability of neurons and reduced blood flow to the area, while repeated high frequency stimulation (>5 Hz) resulted in the opposite effect, with increased neuron excitability and blood flow to the area that was stimulated [90]. The shape of the coil also impacts the course of treatment: a circular coil activates a broader region of the brain, a figure-8 coil can focus on an area that is about 5 mm^3^ in size, and an H-coil can target structures that are up to 6 cm deep [91]. Additionally, TMS can be conducted as a single pulse aimed to stimulate an observable motor output, a paired pulse that stimulates the cortex with 2 pulses separated by a changeable time delay, and repetitive pulse type which involves rapid sequences of magnetic pulses, used for longer lasting modulation [19]. The usage of a specific type of TMS depends on which area of the brain is targeted, where it is located, and if inhibitory or excitatory effects are desired. In the few studies analyzing the direct effects of TMS stimulation on ADHD patients, the side effects of TMS treatment were found to be mild, including headaches and scalp discomfort. However, many of these studies were conducted on small sample sizes, and more research is necessary to fully grasp the scope of applications for this treatment and its side effects on ADHD patients [90].

Transcranial direct stimulation (tDS), also known as transcranial DCS (tDCS), uses multiple externally placed electrodes and direct electric current of 0.5 to 2 mA to the targeted regions of the brain between the negative cathode and positive anode on the scalp [19]. In Figure 3 section D, graph D(i) displays the behavior of the tDCS waveform as a non-phasic direct current. This treatment involves phases of gradually increasing or decreasing the stimulation that can be designed to suit the patient. A trial involving 37 adults demonstrated that continuous treatment resulted in a significant reduction in the clinical manifestations of hyperactivity and inability to focus in the patients with ADHD, with the same mild side effects as previously mentioned [90]. Pilot studies have shown that tDS may be used to improve cognitive performance, including memory and attention, and may show the greatest benefit when conducted with cognitive training [19].

Another promising method of neurostimulation for ADHD treatment is Trigeminal Nerve Stimulation (TNS) which employs either external electrodes and an external, on-body pulse generator (eTNS) or subcutaneously implanted electrodes and an accompanying implantable pulse generator (sTNS) that delivers pulses to the trigeminal nerve and can be programmed to specific charges, frequencies, and time periods for each patient [90]. The sensory root of the trigeminal nerve is located in front of the ear with an ophthalmic branch that extends to the forehead, as shown in Figure 4. The exact mechanism of this treatment, which is currently used for patients with epilepsy or chronic depression, is still under investigation, but it can be hypothesized that the stimulation increases the levels of noradrenaline in the hippocampus and prefrontal cortex [90]. It is unclear if stimulation of the trigeminal nerve activates particular regions of the brain known to be associated with ADHD. One region that may be potentially impacted is the anterior cingulate cortex which is responsible for focusing and is compromised in ADHD patients [93,94]. It also affects the inferior frontal gyrus, which is associated with volume loss in people with ADHD [94,95]. The device is used during sleep, which may be beneficial for patients who have difficulty adhering to such a therapeutic intervention during the day. Two studies investigating TNS for ADHD patients found that it helped inattention symptoms and the parents of the subjects reported improved behavior [90]. More studies need to be conducted with larger trial sizes to confirm the efficacy and safety of these treatments, as well as collect more information on the side effects. Neurostimulation is a new technology, and some parents may be hesitant to subject their child to electrical pulses to the head. However, the different varieties of neurostimulation are used for other conditions, including migraines and seizures, so with more time and research, this may become a widely adopted treatment for patients for whom a pharmaceutical option is not indicated or who desire long-term results from the treatment.

Several companies are targeting this avenue of ADHD therapy to create portable neurostimulation devices for in-clinic and at-home therapy. NeuroSigma was the first to receive FDA clearance, using the de novo premarket review pathway, for their neurostimulation device for pediatric ADHD, called The Monarch^®^ eTNS^®^ System. The device consists of the main component that generates pulses to stimulate the trigeminal nerve and an electrode array accessory to deliver the pulses [97]. It operates using radio frequency energy for over 8 h, but the duration of treatment for each patient is determined by the physician. Currently, there is no clinical evidence supporting any specific timeline, frequency of treatment, or length of treatment. The device can deliver between 0.2 and 10.0 mA at a frequency of 120 Hz. The Monarch^®^ is battery-operated, rechargeable, and involves minimal steps to assemble and use. The kit is sold for around USD 1000, with enough disposable electrode pads for 4 weeks, and additional electrode pads sold for USD 70. The device is worn on the forehead, and treatment occurs when the patient is asleep, so their daily habits are minimally disrupted. However, this may not be ideal for patients with sleep disorders or those whose sleep is accompanied by abnormal movements and physical behaviors, who could peel off the electrodes or otherwise disrupt the system throughout the night. The device does not have a timer, so the patient must sleep for the exact duration of treatment, although no side effects are listed for longer stimulations. This device is designed for children over the age of 7 and cannot be used for any patients with cardiac implants or any other body-worn electronic devices, such as insulin pumps. The instructions are specific, as the pads cannot be placed anywhere on the body besides the forehead and cannot be used in the presence of radio frequency energy, such as that of a television, cell phones, or microwaves due to the radio frequency used by the machine. These guidelines may lead to compliance failure, especially in the pediatric and hyperactive audiences that may find keeping track of these specifics to be complicated. Additionally, the Monarch^®^ requires the patient to set the desired current, so a parent or caregiver plays a heavy role in ensuring the treatment is delivered safely and effectively. The device includes a lock button and an optional input passcode so caregivers can ensure that the patient is not altering the settings without permission. Clinicians need to be familiar with the patient’s family, routine, and domestic life before prescribing, because inadequate oversight may render the treatment ineffective or potentially dangerous for young patients. NeuroSigma^®^ conducted a double-blind, randomized study with 56 children, 30 active participants, and 26 sham participants between the ages of 8 and 12 years old. The patients were diagnosed with the ADHD-RS-IV scale, comprised of 18 questions about the child’s behavior in the past month completed by a parent or guardian [98]. This survey is designed for children and adolescent patients, and reviewers acknowledge internal consistency and reliability for diagnostic criteria [99]. The side effects most prominently reported in the experimental group included trouble sleeping, drowsiness, headaches, stuffy nose, and increased appetite. The patients in the sham group also reported trouble sleeping, drowsiness, stuffy nose, and increased appetite to a lesser extent [100]. The EEG findings demonstrated treatment-related differences in cortical activation and the ADHD-RS and CGI-I scores for the patients receiving active treatment improved. Treatment was discontinued after 4 weeks, and a decline in ADHD-RS scores was observed between 4 and 5 weeks after treatment was discontinued [100]. Detailed statistical analysis of the FDA-approved clinical trial data originating from 34 male children between the ages of 8–12 (mean = 10.3) compared the 25 responders and 26 non-responders [101]. Among responders, TNS treatment was accompanied by significantly increased right frontal EEG power, predictive of improved executive functions and reduced ADHD symptoms. Although this study does not provide data for drawing conclusions about the long-term effects of treatment or treatment for longer periods of time, it does illustrate the safety and efficacy of the treatment as no severe side effects were observed. Based on the improvement in the scores of the experimental group, NeuroSigma^®^ believes this treatment could fill a gap in ADHD treatment and provide safe and longer-lasting treatment. Further studies should investigate the underlying mechanisms of the treatment, provide an explanation for the decline in ADHD-RS scores following treatment, and thoroughly assess any blood pressure or heart rate effects that could be caused by stimulation of the trigeminal nerve.

Other companies, such as Innosphere, have similar products under clinical trial in Europe. Similar to The Monarch^®^, their device, called the AF-RNS^®^, is worn as a cap but has a unique machine-learning capability [102]. Treatment is projected to be applied in ten sessions of only 20 min each per day within a month. Unlike the Monarch^®^, this device is worn during the day and involves parallel cognitive training tasks.

An alternative to the above methods is known as cognitive behavioral therapy (CBT). This method consists of psychoeducation, boosting the motivation of the patient, and traditional organization/planning and adaptive thinking training [103]. CBT sessions are designed to be administered in a long-term setting and are ideally scheduled in such a way that the patient never misses a session for the sake of redundancy and compliance. This model is traditionally used for adults; however, CBT has proven to be adaptable for children to enable them to acquire new ADHD management skills [103]. A new and emerging form of this therapy known as prescription digital therapy, PDT, is in the preliminary phase, with the FDA recently approving EndeavorRx as the first of its kind digital game therapy [104,105]. In a proof of concept study, attention and memory were found to be improved greatly in adolescents with ADHD [106]. Similarly, AKL-T01 is an investigational digital therapeutic delivered through a video game-like interface in at-home play (25 min per day, 5 days per week for 4 weeks). Changes in the mean pre-intervention to post-intervention Test of Variables of Attention (TOVA) Attention Performance Index (API) scores confirmed that AKL-T01 improved attentional performance in pediatric patients with ADHD [107]. Recent reviews have established the growing evidence that DHTs, such as EndeavorRx and AKL-T01, might be used to improve objectively measured inattention in pediatric patients with ADHD while presenting minimal adverse events [20,108].

There is considerable potential for electrostimulation devices to move to at-home care and join the growing family of the Internet of things (IOT) with an intersection with digital or remote health. This provokes consideration of the relationship between aesthetics, as an element of biomedical device design, and in-home adherence to therapy using devices to address ADHD. Research surrounding color perception in ADHD patients is noteworthy. A study comparing the success of individuals with and without ADHD in rapid color identification tests concludes that ADHD patients are more likely to make errors and take longer in tasks that involve quickly identifying and naming color, especially on the blue-yellow color axis [109]. The perception of color is partially controlled by retinal dopaminergic neurons that regulate the system of photoreceptor neurons that convert light into neurological impulses. A deficit of retinal dopamine is associated with a blue-yellow color perception impairment in other neurological disorders that involve altered dopaminergic transmission, including Parkinson’s disease and Tourette syndrome [109]. Contrarily, it was observed that the patients with ADHD did not exhibit a significant difference in color perception compared to their unaffected peers [109]. The aesthetic design of biomedical devices in the context of at-home therapy for ADHD is thus an area worthy of further research.

## 8. Conclusions

ADHD is a complex disorder that must be fully understood to avoid passing unwarranted judgment on people for things out of their control and allow those individuals equal opportunities for success in their education and workplace. As new treatment options are being developed, patients are better able to manage and understand their symptoms. However, patient adherence to these new treatments is crucial, and with the current understanding of the symptoms of this diagnosis, this presents a challenge in itself. While great strides have been made in identifying genetic factors, research is still being conducted to discover all the factors, including environmental factors, that influence the etiology and progression of the condition. As ADHD awareness increases, many preconceived notions are being reevaluated, allowing for these patients to more comfortably seek treatment and discuss their symptoms with their friends and families.

## 9. Future Prospects

In order to further develop neurostimulation as a widely accepted therapy, large clinical trials are needed to delineate the attributes of neurostimulation that address symptoms of ADHD. Studies that aim at the underlying mechanisms of action for electrostimulation devices used in ADHD therapy are necessary to gain public approval for this treatment. With this information, the use of neurostimulation technologies is likely to receive growing attention, mainly towards the unique modulation attributes of frequencies, duty cycle, and duration that are needed to address the classes of symptoms associated with ADHD. There is likely to be rapid migration from the clinic to at-home therapy supported by the Internet of Medical Things (IoMT) and remote health monitoring. The attendant challenge of patient adherence to therapy will necessitate design considerations to promote patient adherence. This includes ergonomic and aesthetic design considerations. Another hurdle presented is increasing the proportion of these patients that receive treatment and ensuring that this technology is convenient and affordable. A non-critical treatment may not be covered by all insurances, further disadvantaging demographics that cannot afford advanced treatments. By increasing the societal awareness of the complexities of ADHD and fighting negative stigma through research and education, more ADHD patients may come to terms with their struggles and seek treatment.

## Figures and Tables

**Figure 1 bioengineering-08-00056-f001:**
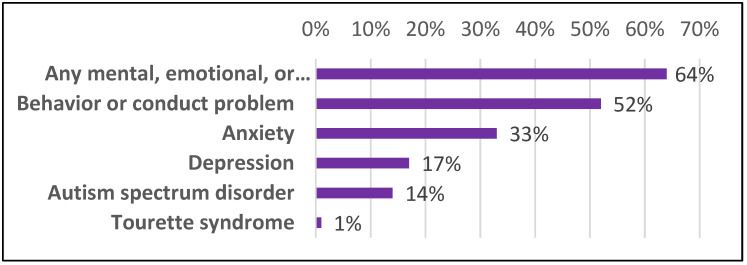
The distribution of mental health comorbidities with an ADHD diagnosis, evidence for a syndromic spectrum of diseases [4,5].

**Figure 2 bioengineering-08-00056-f002:**
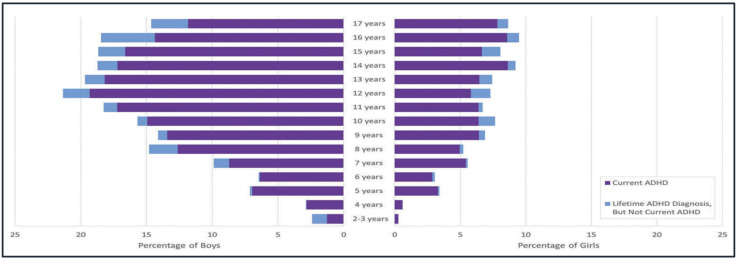
The weighted prevalence estimate distribution of parent-reported ADHD diagnosis by age and gender [5].

**Figure 3 bioengineering-08-00056-f003:**
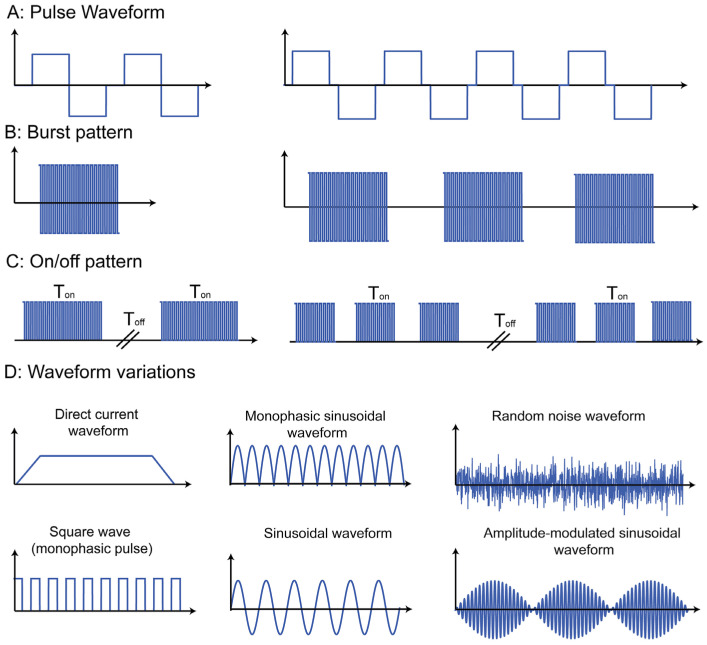
Patterns of transcranial electrical stimulation (tES). (**A**) Pulse waveforms of rectangular waves. (**B**) Pulse waveform applied as a burst. (**C**) Pulse waveforms of increasing frequencies and separated by periods of non-stimulation (ON/OFF). Pulse waveforms of increasing frequencies and separated by periods of non-stimulation (burst). (**D**) Variations in waveforms specified by stimulation parameters, including frequency, pulse shape, pulse width, pulse amplitude, pulse interphase delay, and the pulse repetition frequency. (i) Direct current stimulation (tDCS) and square wave stimulation; (ii) sinusoidal monophasic and biphasic stimulation at a single frequency (tSS); (iii) random noise stimulation (tRNS); and amplitude-modulated sinewave [92].

**Figure 4 bioengineering-08-00056-f004:**
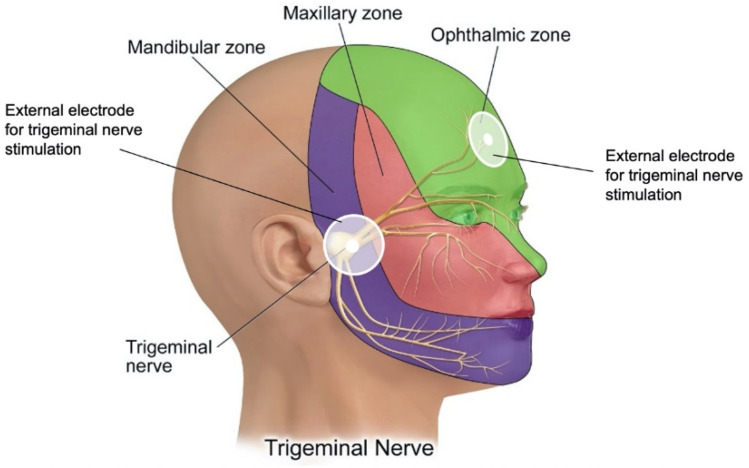
Neurostimulation as exemplified by electrical stimulation of the trigeminal nerve on the temple region and on the forehead region. (Adapted from [96]).

**Table 1 bioengineering-08-00056-t001:** DSM-5 criteria for diagnosis of ADHD in adults [55,56].

Inattention Subtype	Hyperactivity-Impulsive Subtype
(1) Lack of close attention to details, or makes careless mistakes	(1) Restlessness, fidgeting or squirming in seat
(2) Struggle to maintain focus on tasks	(2) Often interrupts others
(3) Pattern of losing necessary objects	(3) Difficulty remaining still, remaining seated
(4) Does not seem to listen when addressed	(4) Often blurts out answers
(5) Difficulty keeping track of assignments/tasks	(5) Struggles to stay quiet, talks excessively
(6) Repeatedly avoids tasks that demand intellectual efforts	(6) Unable to stay on task
(7) Difficulty following through on instructions and/or finishing tasks	(7) Often active when and where it is not appropriate
(8) Difficulty organizing and/or prioritizing tasks and activities	(8) Difficulty with quietly participating in leisure activities
(9) Easily distractible	(9) Feel as though they are “always on the go” or “driven by a motor”
(10) Forgetful with daily tasks and activities	(10) Difficulty with waiting their turn

## Data Availability

Not applicable.

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
