# Peer review of "Understanding ADHD: Toward an Innovative Therapeutic Intervention"

_bioengineering, 2021, doi:10.3390/bioengineering8050056_

Round 1
Reviewer 1 Report
This is a review article on the subject of innovative management of ADHD.
The article traces the epidemiology, evaluation and treatment of drugs and others, and especially neurostimulation.
Regarding the criteria for evaluating the disorder, the WHO criteria and scales such as ASRS 18 or the Wender Utah Rating Scale could have been mentioned.
What is the point of citing the work of anthroposophical therapy? no evidence based medicine
Why not have approached the work in CBT, very numerous and effective both in children and adults?
Minor revisions
Reviewer 2 Report
Line 44: When the diagnostic process is already mentioned here, it should be described in much more detail. ADHD is not just diagnosed by medical professionals, also by psychologists.
Line 54: It is unclear on which basis the eTNS was approved as a therapeutic intervention. Were randomized trials performed? Effect sizes? Emotions were mentioned as a therapeutic target. Therefore, emotional dysregulation should be mentioned as a primary symptom.
There is no reference to and no discussion in the text regarding Figure 1.
There is no reference to and no discussion in the text regarding Figure 2.
The paragraphs on Serotonin and Dopamine seem misplaced and statements like „Enhancements or stabilization of serotonin levels may be the outcome of neurostimulation“ are highly speculative without any reference.
Part 3 Clinical Manifestations: Nothing is said about the high persistence of the disorder into adulthood as just childhood aspects are covered.
Part 4 Diagnosis: It is unclear why the Conners scales and the Qb(+) Test are not mentioned.
There is no reference to Table 1 in the text.
The mentioned studies regarding EEG diagnosis in ADHD are not convincing to me (54,55) as the authors state the unreliability of these measures.
Clinical management: Nothing is said about the evidence base of the therapeutic interventions. This is essential when giving an overview over interventions. The reference to anthroposophical therapy (line 366) is unclear as article 71 is a non-control group design and giving therefore a weak empirical evidence.
Part 7: The so-called innovations in therapeutic intervention which are mentioned in the title of this manuscript cover only a relatively short part. They seem to be in a very preliminary stage as in reference 91 one can see that just a medium effect with questionable clinical relevance was achieved.
The scope of the manuscript is unclear to me. At first, it looked like some kind of educational review on ADHD leaving out several important aspects. The title does not fit with the content of the manuscript. Results in neurological stimulation in ADHD are very preliminary and need much more empirical support, also in adults, before this area can be labeled as a new innovative therapy.
Less...
Reviewer 3 Report
The manuscript narratively reviews the etiopathogenesis of ADHD and ADHD treatments and suggests the treatment of non-invasive brain stimulation treatment in ADHD. The topic is clearly of interest, however, I have several issues that should be modified before publishing.
- Abstract: ADHD is not in every case a progressive disease, so disease progession is a bit misleading
- Page 2. line 47: better pharmacological and/or non-pharmacological therapeutic interventions (instead of psychiatric/psychological)
- Which part of the brain that are associated with ADHD are stimulated by trigeminal nerve stimulation?
- I do not understand why there is a whole and detailled paragraph about LPHN3? This is only one of the many risk genes associated with ADHD and I do not see a special connection to non-invasive brain stimulation therapies here?
- Page 4, line 126: Why emphasize on serotonin? ADHD is potentially much more closely related to dopamine and norepinephrine dysbalances. Norepinephrine is missing here.
- Page 4, line 131 What means normal range of dopamine? In blood? In cerebrospinal fluid? In the brain?
- Page 4, line 146: after DSM-5, only 5 or more symptoms are needed, not 6 (that was DSM-IV)
- Page 5, line 88: Sleeping problems truly are often associated with ADHD and sometimes ADHD medication can disturb sleep, however ADHD medication is not making ADHD worse, this is how the sentence sounds like...
- Page 5, line 204: which study? The manuscript is a review, not original research.
- Page 8, line 318: Is there really a transdermal MPH application available?
- ADHD medications missing: Guanfacine, Lisdexamfetamine
- Page 8, line 334: Opioids work very differently from stimulants, I do not understand that analogy here.
- The authors mention anthroposophical therapy but there is no evidence for that in ADHD treatment. Why do they not mention the first line and evidence based psychotherapy interventions that are also part of each guideline: parent training (children) and cognitive behavioural therapy (adults). There are also frist studies giving hints at the efficacy of mindful ness based interventions and physical activites.
- What about neurofeedback? The efficacy is lower than of the stimulant medication but it should at least be mentioned in the treatment options. They also cite a study about neurofeedback, but do not discuss it in the manuscript text...
- Page 12, line 499: what does misuse in this context mean?
- Overall, I would suggest to shorten the paragraphs on etiology and pathogenesis and pharmacological treatment of ADHD and discuss the non pharmacoological treatment options including the focus of the review on neurostimulation in more detail.
In the whole manuscript, there are no references in the main text to the different figures and tables.
Intermittendly, whole sentences are in italics, I think this might not be done on purpose, e.g. see page 10, line 315, 320, 326.
Reviewer 4 Report
Dear Authors,
Thank you for your contribution. I recommend following suggestions to reconsider your manuscript. Please answer pointwise and incorporate all the changes, discussion, and suggested references in suitable places for us to be clear about the revised/edited manuscript. Thank you for your contribution.
Thank you for your contribution, this is interesting study.
- Can we discuss about “newly developed adaptive frequency-based model” and stratification of patients for neurostimulation therapy. A recent paper on this as below: https://pubmed.ncbi.nlm.nih.gov/33577937/
- Wish if authors could expand discussion on digital therapy a little more. A lot has been going on and primary and secondary outcomes using Test of Variables of Attention (TOVA). This is associated with no side effects so far. Refer to following: https://pubmed.ncbi.nlm.nih.gov/33334505/
https://pubmed.ncbi.nlm.nih.gov/33515870/
- Can you check please if Figure 4, you have taken permission from the authors or the relevant authorities. I am not sure just writing adapted from them in the legend is sufficient. Please check, I am sorry I am not very familiar with this rule but thought to bring to your notice.
- Authors have discussed Neuropsychiatric EEG-Based Assessment Aid (NEBA) system for supporting a clinical ADHD diagnosis. They should also add the Electroencephalographic Predictors of Treatment Response. Please refer to https://pubmed.ncbi.nlm.nih.gov/33068751/

Author Response
Response to Reviewers Comments
Manuscript ID: bioengineering-996332
Understanding ADHD: Toward an Innovative Therapeutic Intervention
Allyson Camp 1, Amanda Pastrano 1, Valeria Gomez 1, Kathleen Stephenson 1, Brianna Perez 1, Hunter Syas 1 and Anthony Guiseppi-Elie 1,2,3,4,*
Dear Authors,
Thank you for your contribution. I recommend following suggestions to reconsider your manuscript. Please answer pointwise and incorporate all the changes, discussion, and suggested references in suitable places for us to be clear about the revised/edited manuscript. Thank you for your contribution.
Thank you for your contribution, this is interesting study.
- Can we discuss about “newly developed adaptive frequency-based model” and stratification of patients for neurostimulation therapy. A recent paper on this as below: https://pubmed.ncbi.nlm.nih.gov/33577937/
We thank the reviewer for bringing our attention to this recent article that was published after the submission of our review. We have expanded our discussion of neurostimulation to include newly developed adaptive frequency-based model and stratification of patients for neurostimulation therapy and have incorporated this recently published citation in our review.
- Wish if authors could expand discussion on digital therapy a little more. A lot has been going on and primary and secondary outcomes using Test of Variables of Attention (TOVA). This is associated with no side effects so far. Refer to following: https://pubmed.ncbi.nlm.nih.gov/33334505/
https://pubmed.ncbi.nlm.nih.gov/33515870/
We are grateful for this reviewer’s embrace of “digital therapy”. All reviewers do not share the same openness to novelty and progress. We have expanded upon our treatment of the newly emerging digital therapies and included, among others, the above citations.
- Can you check please if Figure 4, you have taken permission from the authors or the relevant authorities. I am not sure just writing adapted from them in the legend is sufficient. Please check, I am sorry I am not very familiar with this rule but thought to bring to your notice.
Thank you for raising this issue and for seeking clarification. It is the acknowledged approach in peer reviewed scientific publications to secure permission when using an original image or data for republication in one’s own work. However, when the original image is being created by the author (us, in this case) but is based on a previously published image, no such permission is necessary. However, it is a common courtesy to say “adapted from” as a way of acknowledging the creative talents of the first author and acknowledge the source of the component of the final image. We believe it is appropriate to state “adapted from” and provide the citation, the way we have done.
- Authors have discussed Neuropsychiatric EEG-Based Assessment Aid (NEBA) system for supporting a clinical ADHD diagnosis. They should also add the Electroencephalographic Predictors of Treatment Response. Please refer to https://pubmed.ncbi.nlm.nih.gov/33068751/
We are grateful for this reviewer’s comment on EEG-Based assessment. We have expanded upon our treatment by including the Electroencephalographic Predictors of Treatment Response and including the citation.

Round 2
Reviewer 2 Report
Diagnosis
I still cannot find any reference to the Qb+ test.
Author´s reply:
„We have now been explicit in stating that EEG is unreliable as a diagnostic tool for ADHD.“
I cannot find this explicit statement in the revised version. The authors of (54, 55) explicitly state the unreliability of these measures.
I still do not agree with the reference to anthroposophical therapy. The positive outcome might have resulted from the weak design. Effects were low, the concept is debatable, so I do not see the point to mention it.
I can only reiterate my first criticism:
The scope of the manuscript is unclear to me. At first, it looked like some kind of educational review on ADHD leaving out several important aspects. The title does not fit with the content of the manuscript. Results in neurological stimulation in ADHD are very preliminary and need much more empirical support, also in adults, before this area can be labeled as a new innovative therapy. To me it is dubious why the FDA already gave a premarket approval to the Monarch eTNS system with such less data. I have a solid background in evidence based medicine and do not see the basis for this decision.
Author Response
Response to Reviewers Comments
Manuscript ID: bioengineering-996332
Understanding ADHD: Toward an Innovative Therapeutic Intervention
Allyson Camp 1, Amanda Pastrano 1, Valeria Gomez 1, Kathleen Stephenson 1, Brianna Perez 1, Hunter Syas 1 and Anthony Guiseppi-Elie 1,2,3,4,*
Author’s Overall Response
I write to strongly recommend that this manuscript and the reviewers’ comments be given to arbitration by another member of the Editorial Board of Bioengineering.
I do so because I strongly disagree with the reviewers’ assessment and more importantly, I strongly disagree with the premise, the reasons and the rationales of this single reviewer’s position.
I point out that two other reviewers have given this paper their full endorsement.
The Author’s Response – Reviewer 2
Diagnosis
I still cannot find any reference to the Qb+ test.
Thank you for this comment. A description of the Conners scales (line 308) and Qb(+) test (line 290) have been added to this section. “Computerized QB Test Plus (Qb+) emphasizes objectivity, allows for remote diagnoses and accommodates independent partitioning among the core symptoms of, hyperactivity, inattention, and impulsivity[43, 44].”
Author´s reply:
„We have now been explicit in stating that EEG is unreliable as a diagnostic tool for ADHD.“
I cannot find this explicit statement in the revised version. The authors of (54, 55) explicitly state the unreliability of these measures.
The authors point our clearly and explicitly on page 10, “As per FDA regulations, this device is not meant to be used as a stand-alone diagnostic technique…”
I still do not agree with the reference to anthroposophical therapy. The positive outcome might have resulted from the weak design. Effects were low, the concept is debatable, so I do not see the point to mention it.
The reviewer is not entitled to “agree” with a cited paper and then speculate “that the positive outcome might have resulted from a weak design”. As authors, we are ethically responsible for all of the published literature. We are required to be critical, but not be neglectful, which is what this reviewer is asking of the authors - to neglect previously peer-reviewed and published works – because the reviewer “do not agree with the reference” and “speculates” about the quality of the study. The study is aptly and measurably critiqued by these authors but cannot be ignored, and will not be ignored, as suggested by the reviewer.
Moreover, the reviewer is not invited to be a co-author of the paper. The authors of this review have an ethical responsibility to address all relevant published works, not just those that one single reviewer “agrees with” and neglect those that the reviewer “disagrees with”. This is not the way of science.
I can only reiterate my first criticism:
The scope of the manuscript is unclear to me. At first, it looked like some kind of educational review on ADHD leaving out several important aspects. The title does not fit with the content of the manuscript. Results in neurological stimulation in ADHD are very preliminary and need much more empirical support, also in adults, before this area can be labeled as a new innovative therapy. To me it is dubious why the FDA already gave a premarket approval to the Monarch eTNS system with such less data. I have a solid background in evidence based medicine and do not see the basis for this decision.
Regretfully, this is not how peer-review is expected to work. The reviewer cannot suffer professional denial because he/she believes themselves a better expert than the experts at the US Food and Drug Administration. This review is clear – the US Food and Drug Administration has provided premarket approval for the marketing of Monarch eTNS system. This is a verifiable fact and the citations are clearly provided to support this fact. The authors of this paper are responsible for reviewing the FACTS. This is the first of its type application of such a device, making it innovative. It is approved ONLY for children, not for adults. Why is it necessary that for it to be defined as innovative it must also address adults? Why is it necessary that for it to be defined as innovative it must be widely used? This is not properly reasoned or logical and may reflect an inadequate appreciation of the word or concept of innovation.
Overall, these few reviewer’s comments are not logical and do not comply with the well-established norms of proper peer review.
The Author’s Response – Reviewer 3
Figure 1, 2 and 3 should be situated closer to the text parts that are related to them to make it easier to read.
Per the reviewer’s suggestion, the Figures 1, 2 and 3 have been moved to be closer to their first mention within the text.
Reviewer 3 Report
The manuscript is much improved, I have only a minor comment:
- Figure 1, 2 and 3 should be situated closer to the text parts that are related to them to make it easier to read.
Author Response
Figures 1, 2 and 3 should be situated closer to the text parts that are related to them to make it easier to read.
Figures 1, 2 and 3 have been relocated within the manuscript so that they are closer to the point of first mention.
Reviewer 4 Report
Revision appropriately done